# Autologous Adipose-Derived Mesenchymal Stem Cells Combined with Shockwave Therapy Synergistically Ameliorates the Osteoarthritic Pathological Factors in Knee Joint

**DOI:** 10.3390/ph14040318

**Published:** 2021-04-01

**Authors:** Jai-Hong Cheng, Ke-Tien Yen, Wen-Yi Chou, Shun-Wun Jhan, Shan-Ling Hsu, Jih-Yang Ko, Ching-Jen Wang, Chun-En Aurea Kuo, Szu-Ying Wu, Tsai-Chin Hsu, Chieh-Cheng Hsu

**Affiliations:** 1Center for Shockwave Medicine and Tissue Engineering, Kaohsiung Chang Gung Memorial Hospital and Chang Gung University College of Medicine, Kaohsiung 833, Taiwan; murraychou@yahoo.com.tw (W.-Y.C.); b9502077@cgmh.org.tw (S.-W.J.); hsishanlin@yahoo.com.tw (S.-L.H.); kojy@cgmh.org.tw (J.-Y.K.); cjwang1211@gmail.com (C.-J.W.); tsaichin1219@gmail.com (T.-C.H.); 2Medical Research, Kaohsiung Chang Gung Memorial Hospital and Chang Gung University College of Medicine, Kaohsiung 833, Taiwan; 3Department of Leisure and Sports Management, Cheng Shiu University, Kaohsiung 833, Taiwan; ktyen@gcloud.csu.edu.tw; 4Department of Orthopedic Surgery, Sports Medicine, Kaohsiung Chang Gung Memorial Hospital and Chang Gung University College of Medicine, Kaohsiung 833, Taiwan; 5School of Nursing, Fooyin University, Kaohsiung 831, Taiwan; 6Department of Chinese Medicine, Kaohsiung Chang Gung Memorial Hospital and Chang Gung University College of Medicine, Kaohsiung 833401, Taiwan; lecherries@gmail.com (C.-E.A.K.); rickywu0818@gmail.com (S.-Y.W.)

**Keywords:** adipose-derived mesenchymal stem cells, shockwave, osteoarthritis, extracellular matrix factors, bone morphogenetic protein

## Abstract

Adipose-derived mesenchymal stem cells (ADSCs) and shockwave (SW) therapy have been shown to exert a chondroprotective effect for osteoarthritis (OA). The results of this study demonstrated that autologous ADSCs had dose-dependent and synergistic effects with SW therapy (0.25 mJ/mm^2^ with 800 impulses) in OA rat knee joint. Autologous, high-dose 2 × 10^6^ ADSCs (ADSC2 group) combined with SW therapy significantly increased the bone volume, trabecular thickness, and trabecular number among in the treatment groups. ADSC2 combined with SW therapy significantly reduced the synovitis score and OARSI score in comparison with other treatments. In the analysis of inflammation-induced extracellular matrix factors of the articular cartilage in OA, the results displayed that ADSC2 combined with SW therapy had a greater than other treatments in terms of reducing tumor necrosis factor-inducible gene (TSG)-6 and proteoglycan (PRG)-4, in addition to increasing tissue inhibitor matrix metalloproteinase (TIMP)-1 and type II collagen. Furthermore, ADSC2 combined with SW therapy significantly reduced the expression of inflammation-induced bone morphogenetic protein (BMP)-2 and BMP-6. Therefore, the results demonstrated that ADSC2 combined with SW therapy had a synergistic effect to ameliorate osteoarthritic pathological factors in OA joints.

## 1. Introduction

Adipose tissue is a widely-used source for isolation of adipose-derived mesenchymal stem cells (ADSCs) for use in regenerative medicine, which has been demonstrated in animal and clinical trials related to various diseases [1]. Adipose tissue contains mesenchymal stem cells at level of 500-fold that of the same volume of bone marrow [2]. Many cell and animal studies have shown that ADSCs have the ability to achieve osteogenic differentiation [3]. In addition, the different types of tissue have been reported to be differentiated by ADSCs, including bone, cartilage, muscle, ligament, tendon, and fat [4]. ADSCs have the paracrine effects in terms of expressing a variety of different growth factors, signaling factors, exosomes, and micro-RNA to facilitate repair and healing of damaged tissue [5,6]. ADSCs also secrete angiogenic factors and can promote the growth of new blood vessels. Many studies have investigated therapeutic strategies to ascertain whether autologous ADMSCs induce biological effects in the regeneration and protection of articular cartilage in osteoarthritis (OA) [7]. Clinical studies have progressed to the use of ADSCs for the treatment of various diseases, particularly in COVID-19 patients with pneumonia, in order to improve clinical outcomes and efficiency [6,8]. However, long-term follow-up is required to elucidate the side effects after treatment with ADSCs. ADSCs also play a role in inflammatory modulation of pro-inflammatory cytokines, including interleukin-1β (IL-1β), IL-6, tumor necrosis factor-α (TNF-α), and interferon gamma in OA [9].

OA is a degenerative cartilage disease that occurs due to inflammation and causes loss of articular cartilage and subchondral bone [10]. Many factors increase the risk of OA, including age, obesity, sport injury, and gender [11], with age and obesity being the main risk factors for OA, especially of the knee. During the process of OA formation, synovitis is induced at an early stage, and lymphocytes, neutrophils, and macrophages infiltrate into the synovial membrane, causing a lining cell layer and stimulation of the release of many cytokines, chemokines, and growth factors [12]. Consequently, matrix metalloproteinases and aggrecanase are induced to destroy the composition of the articular cartilage, causing losses of integrity, and sclerosis, bone cysts, and osteophytes are then formed in the subchondral bone [10]. In an OA knee joint, pro-inflammatory cytokines IL-1β and TNF-α are induced, which affect catabolism of the cartilage extracellular matrix and anabolism of growth factors [13]. Both of these cytokines break down the balanced regulation of catabolic proteases such as aggrecanases and collagenases, as well as anabolic growth factors such as TGF-β and bone morphogenetic proteins in the inflammatory processes in OA cartilage [13,14,15,16]. In addition, white adipose tissue present in obesity releases adipokines, which can stimulate inflammatory cytokines to enhance the expression of metalloproteinases, aggrecanase, and NOS2 [17]. Adipokines such as leptin, TNF-α, and IL-6 have been reported to have biphasic effects in the articular cartilage homeostasis and cause destruction of OA [17]. Therefore, the imbalance of catabolic factors and anabolic factors has a direct effect on the destruction of cartilage in OA.

ADSCs and shockwave (SW) therapy have been demonstrated to exert chondroprotective effect to decrease cartilage degradation and improve subchondral bone remodeling in the treatment of OA [18,19]. SW therapy is a physical, non-invasive, and safe method by which to repair the pathological damage that occurs in the musculoskeletal disorders [20]. The effects of SW therapy on tissue regeneration act as a mechanical stimulus, which triggers angiogenesis, immunomodulation, chondroprotection, bone remodeling, and stem cell homing [20,21,22]. All the detailed biological effects of SW are exerted through mechanotransduction to interact directly with cells in tissue and regulate the expressions of growth factors, signaling factors, transcription factors, cytokines, and chemokines [20,23]. Many cellular signaling transductions have been reported to occur via the mechanotransduction caused by SW therapy, including the focal adhesion kinase (FAK), extracellular-signal-regulated kinase (ERK), Wnt/β-catenin, ATP/P2X7, and toll-like receptor 3 (TLR3) signaling pathways [23,24]. SW therapy has been proposed to be of significant potential for use in translational medicine and alternative therapy, or even in combination therapy for the treatment of a broad range of diseases [25,26].

In the current study, we compared two concentrations of autologous ADSCs for the treatment of OA in a rat model. In addition, transplantation of high-dose autologous ADSCs combined with SW therapy was performed, which could synergistically ameliorate the process of inflammatory changes in the pathological deterioration of OA knees. The combination therapy was shown to contribute a greater benefit than the individual treatment along in knee OA [25,27]. Compilation of milestone discoveries related to autologous ADSCs combined with SW therapy should aid in further advancement in the field of cell biology and therapy towards clinical applications. Finally, the results of the current study provide more information and technological knowledge in relation to ADSCs combined with SW therapy for OA of the knee.

## 2. Results

### 2.1. Study Design and Identification of Autologous Rat ADSCs

To observe the combination effects of pathological changes caused by treatment with autologous ADSCs and SW therapy for OA, doses-dependent assays of autologous ADSCs, SW treatment along, and combination therapy were performed, and the results compared (Figure 1A).

The aim of the first experiment was to identify autologous rat ADSCs by morphological observation and characterization of specific cell-surface markers (Figure 1B). The ADSCs displayed the classic spindle-shaped morphology and fibroblast-like structure of mesenchymal stem cells (Figure 1B). The autologous ADSCs of rat were stained with antibodies to against typical mesenchymal markers, including CD29 (99.99%) and CD90 (96.23%) (Figure 1B), and the negative markers of CD45, CD106, RT1a, and RT1b. Flow cytometry was also performed for analysis.

### 2.2. Dose-Dependent of Autologous ADSCs and SW Combination Therapy Improved Bone Remodeling in the Treatment of Knee OA

In line with our previous study, autologous ADSCs were precisely injected intra-articularly into the rat knee using real-time ultrasound guidance (Appendix A). Thereafter, SW (0.25 mJ/mm^2^, 800 impulses) was applied to the medial tibia of the rat knee to activate autologous ADSCs at the seven weeks for knee OA treatment (Figure 1A and Appendix A).

Micro-CT images displayed severe erosions of subchondral bone in the OA group and were improved in the treatment groups (Figure 2A). The dose-dependent effect of autologous ADSC was demonstrated by comparing the autologous ADSC1 (1 × 10^6^ cells) and ADSC2 (2 × 10^6^ cells) treatment groups. Further, the combination therapy provided in the ADSC2 + SW group was the most successful treatment, as compared with the SW, ADSC1, ADSC2, and ADSC1 + SW groups.

Micro-CT scan data were analyzed to illustrate the subchondral bone volume, trabecular thickness, and trabecular number, all of which were significantly improved in the region of interest (ROI) of the proximal tibia among the treatment groups (Figure 2A,B). The trabecular bone volume fraction (BV/TV), trabecular thickness, and trabecular number were most significantly improved in the ADSC2 group, followed by the ADSC1 group (61.21 ± 0.274, 0.202 ± 0.004, and 3.149 ± 0.050 versus 59.83 ± 0.639, 0.198 ± 0.003, and 3.025 ± 0.048), and finally the ADSC2 + SW group then the ADSC1 + SW group (62.00 ± 0.605, 0.204 ± 0.003, and 3.201 ± 0.018 versus 61.46 ± 0.280, 0.195 ± 0.002, and 3.036 ± 0.112, respectively). In addition, ADSC2 + SW acted more effectively and synergistically in terms of improvements in the BV/TV value, trabecular thickness, and trabecular number than other treatment groups (*p* < 0.05 and *p* < 0.01) (Figure 2B). The results demonstrated that high-dose ADSCs plus SW therapy improved bone remodeling in the knee OA.

### 2.3. Autologous ADSCs Combined with SW Therapy Significantly Reduced the Destruction of Articular Cartilage and Synovitis in Knee OA

Autologous ADSCs combined with SW therapy protected against damage to the articular cartilage for the treatment of knee OA. The pathological change of hyaline cartilage was shown by safranin-O staining in the Sham, OA, ADSC1, ADSC2, ADSC1 + SW, and ADSC2 + SW groups (Figure 3A, left). The level of protective effect on the articular cartilage was measured using OARSI score, which showed that ADSC2 treatment was more effective than ADSC1 treatment (1.50 ± 0.528 versus 7.37 ± 0.498, respectively, *p* < 0.001) and ADSC2 + SW treatment resulted in the best recovery score of all treatments as compared with OA group (*p* < 0.001) (Figure 3A, right) (Appendix A). Our results indicated that autologous ADSC2 + SW treatment had the best synergistic effect for protection of the articular cartilage in knee OA.

Synovitis of the synovial membrane was measured after treatment in each group (Figure 3B). Mononuclear cells and neutrophils were observed to infiltrate the synovial membrane in knee OA. Lining cell layers were observed to be composed of large cells, forming an irregular layered tissue (Figure 3B, left). Synovitis improved significantly in all treatment groups as compared with the OA group (*p* < 0.01 and *p* < 0.001). Among the treatment groups, the ADSC2 + SW group exhibited the best outcome of all groups with regards to synovitis in knee OA (*p* < 0.05 and *p* < 0.001).

### 2.4. Specific Extracellular Matrix Factors Were Modulated after Autologous ADSCs and Combination Treatments in the Inflammatory Knee OA

The extracellular matrix proteins are significant components of the articular cartilage of knee. In the current study, TSG-6 and PRG-4 were found to be highly expressed throughout the hyaline cartilage in the OA group (Figure 4A,B), and were significantly reduced to normal levels after treatment in comparison with the Sham group (*p* < 0.05 and *p* < 0.001). Notably, the expression of TSG-6 and PRG-4 in the deep regions of OA cartilage were very significantly reduced after SW, ADSC1, ADSC2, ADSC1 + SW, and ADSC2 + SW treatment in comparison with the OA group (*p* < 0.001) (Figure 4A,B, triangle: deep region; arrowhead: cartilage surface). Finally, ADSC2 + SW treatment was the most effective treatment among all treatment groups (*p* < 0.001), and reduced inflammation-induced TSG-6 and PRG-4 proteins to normal levels in comparison with the Sham group.

Subsequently, TIMP-1 and type II collagen were found to be significantly reduced in the OA group (*p* < 0.001) and increased in the treatment groups (Figure 4C,D. triangle: deep region; arrowhead: cartilage surface). Among the treatment groups, ADSC2 + SW (*p* < 0.05) had a synergistic effect in promoting the expression of TIMP-1 and type II collagen to repair the damage to the articular cartilage.

### 2.5. Effect of Autologous ADSCs Combined with SW Therapy in Regulation Inflammation-Induced BMP-2 and BMP-6 in the Treatment of Knee OA

The key growth factors of BMP-2 and BMP-6 have been reported to have dual roles in normal and osteoarthritic cartilage. Immunohistochemical analysis of BMP-2 and BMP-6 in the articular cartilage of the Sham, OA, ADSC1, ADSC2, ADSC1 + SW, and ADSC2 + SW groups was performed, as shown in Figure 5. The expression of inflammation-induced BMP-2 was decreased in all treatment groups as compared with OA group, which had a high expression (*p* < 0.05, 0.01 and 0.001) (Figure 5A, triangle: deep region; arrowhead: cartilage surface). Among the treatment groups, the ADMSC2 and ADMSC2 + SW groups showed the best effects in terms of returning the BMP-2 expression to a normal level as compared with the Sham group (*p* < 0.05 and *p* < 0.001). In addition, inflammation-induced BMP-6 was significantly reduced in the ADSC2 + SW group as compared with other groups (*p* < 0.05) (Figure 5B). The results demonstrated that ADSC2 + SW synergistically modulated the high expressions of inflammation-induced BMP-2 and BMP-6 growth factors in knee OA (*p* < 0.05).

## 3. Discussion

In the current study, ADSCs treatment for knee OA exhibited a dose-dependent effect and a synergistic action with SW therapy in the protection of the articular cartilage with inflammation. High-dose autologous ADSCs combined with SW therapy significantly improved the destruction of the articular cartilage and subchondral bone and reduced synovitis. Our data showed that the expression of inflammation-induced TSG-6, PRG-4, BMP-2, and BMP-6 were reduced, while those of matrix metalloproteinases regulator, TIMP-1, and type II collagen were enhanced by ADSC2 + SW treatment for knee OA in rats. The results of the study demonstrated regulation of anabolic factors and inflammation-induced growth factors by ADSCs and SW therapy individually, or ADSCs combined with SW therapy, in the articular cartilage of OA rat knees. All results were summarized in the Figure 6.

ADSCs and SW therapy have been reported to promote bone remodeling [28,29]. ADSCs has a broad capacity to differentiate specialized cell types including neural, cartilage, bone, and others [28]. ADSCs could release particular factors to promote bone formation, including vascular endothelial growth factor (VEGF), bone morphogenic protein 2 (BMP2), runt-related transcription factor 2 (RUXN2), and alkaline phosphatase (ALP). In addition, SW therapy produces mechanical force to stimulate bone tissue to release several osteogenic factors, including VEGF, ALP, BMP-2, RUNX2, osteocalcin, and IGF1 in bone healing [30]. Further, SW therapy has been reported to promote mesenchymal stem cells (MSCs) proliferation and differentiation for tissue repairing [31]. All the evidence suggests that ADSCs combined with SW therapy may have the synergistic effects in the tissue regeneration [25]. In our study, the micro-CT analysis displayed that high-dose ADSCs and combined with SW therapy promote the bone healing more efficiently than other treatments (Figure 2). However, the detail mechanism of the combination therapy need to further investigation in future.

Tumor necrosis factor (TNF)-α-stimulated protein 6 (TSG-6) is a secreted protein that was originally first cloned from TNF-treated human fibroblasts [32]. As the name of this protein implies, it is stimulated by pro-inflammatory cytokines TNF-α and IL-1, and has been reported to be expressed in inflammatory tissues and a variety of diseases in addition to OA [33]. TSG-6 directly interacts with glycosaminoglycan hyaluronan and is induced in inflammatory disease regions such as synovial fluids and articular chondrocytes in rheumatoid arthritis and OA [33,34]. TSG-6 has dual functions in terms of a positive effect of chondroprotection and a negative effect of OA progression [33,35]. In the current experiment, TSG-6 was highly-induced throughout the articular cartilage in the OA group (Figure 4A, triangle and arrowhead). After treatment, the ADSC1 and ADSC2 groups exhibited a low expression of TSG-6 throughout the degenerative cartilage, but the SW, ADSC1 + SW, and ADSC2 + SW groups were almost at cartilage surfaces, with a normal TSG-6 expression (Figure 4A, arrowhead). The results indicated that SW therapy may help to inhibit overexpression of TSG-6 in the deep degenerative cartilage. It has been reported that TSG-6 expression in the deep OA cartilage could inhibit its matrix synthesis and assembly to increase the risk of OA [33]. ASDCs and SW therapy are reported to have the immunomodulation to regulate the expression of TNF-α and IL-1 [20,36]. This indicated that both treatments could modulate the expression level of TSG-6 in OA cartilage. Consequently, normal level of TSG-6 on the surface of cartilage may interact with inter-alpha inhibitor to improve hyaluronan–aggrecan assembly and promote cartilage regeneration [33,37]. However, the mechanism of ADSCs and SW therapy in modulation of TSG-6 to improve OA progression remains unclear and requires further evaluation.

Proteoglycan 4 (PRG-4), also called lubricin and superficial zone protein, is a proteoglycan that is expressed in synovial lining cells, synovial fluid, and on the surface of articular cartilage (Figure 4B, Sham group, left image) [38]. In the severe OA (after six weeks of OA development), in this study, PRG-4 was induced to a level over 2.5-fold that in the Sham group (Figure 4B, right panel). The distribution of PRG-4 was a high expression in the deep and surface-damaged regions of the articular cartilage in the OA group and reduction to a normal level in the ADSC2 and ADSC2 + SW groups as compared with the Sham group (Figure 4A, right panel). Many reports have indicated that PRG-4 is reduced in OA progression, but a high expression improves cartilage regeneration [39,40]. Therefore, the results of this study may indicate that PSG-4 is increased during OA progression to promote cartilage regeneration [41]. ADSCs and SW therapy modulated the anabolism of the ECM and effected recovery of the articular cartilage of knee joint in this research.

The BMP superfamily has important dual roles, with a function in cartilage protection and harmful effects on cartilage repair during OA progression [42]. In this study, the expression profiles before and after treatment for OA of BMP-2 and BMP-6 were surveyed. BMP-2 is presented in normal articular cartilage; however, it is also detected in osteoarthritic cartilage, in both clustered and individual chondrocytes [43]. In mouse models of OA, the results showed strong up-regulation of BMP-2 in the cartilage, and in addition, IL-1 and TNF-alpha stimulated cartilage cells [44]. The expression of BMP-2 in normal and osteoarthritic cartilage has a significant role in tissue homeostasis and maintenance of tissue integrity during the progression of OA [16]. In addition, BMP-6 has been reported to be expressed with no difference in healthy and OA chondrocytes [45]. Further, as per the function of BMP-2 in OA, BMP-6 can stimulate total proteoglycans synthesis, and is also involved in homeostasis and in the maintenance of joint integrity in OA [45]. Our results showed that BMP-2 and BMP-6 were significantly increased in the OA group and were modulated by ADSCs and SW therapy during cartilage regeneration (Figure 5). Indeed, ADSCs combined with SW therapy was more efficient in terms of cartilage protection, and was demonstrated to reduction of inflammation-induced BMP-2 and BMP-6 in OA than other treatments.

The limitations of this study were as follows. The study employed autologous ADSCs from rodents, and the therapeutic outcomes may be difference to those of large-animal or human clinical trials. Severe OA was created at six weeks post-surgery, and then treatment with ADSCs, SW therapy, and combination therapy then commenced. The pathological changes and recovery may be dependent on the timing of treatment and the number of injected cells. Different types of shockwave device may generate a different mechanical force or mechanotrasduction with various energy levels and impulses to induce biological effects on tissue [24]. The optimal doses of ADSCs, SW therapy, and combination therapy used in animal studies may not be suitable for translation for use in human clinical trials. Autologous ADSCs were sourced from rats; the experimental protocol may not be suitable for cell preparation in human clinical trials.

## 4. Materials and Methods

### 4.1. Experimental Rats

All rats (42 eight-week-old Sprague–Dawley rats) were treated humanely within the guidelines of the Guide for the Care and Use of Laboratory Animals of the National Institute of Health. All animals were housed at 23 ± 1 °C with a 12-h light and dark cycle, and were given food and water. The Center for Laboratory Animals at Kaohsiung Chang Gung Memorial Hospital (KCGMH) provided veterinary care to the rats. The study was approved by the Institutional Animal Care and Use Committee (IACUC) at KCGMH. The experimental design is shown in Figure 1A.

### 4.2. OA Rat Model

The left knee of the individual rat was prepared in surgically sterile fashion. An OA model was created by anterior cruciate ligament transection (ACLT) and medial meniscectomy (MMx) with a 1:1 volume mixture of Rompun (5 mg/kg) (xylazine–hydrochloride, Bayer, Leverkusen, Germany) and Zoletil (20 mg/kg) (Tiletamine–zolazepam, Virbac, Carros, France). The knee joint was irrigated, and the incision was closed. Prophylactic antibiotics with ampicillin 50 mg/Kg every 6 h were administered for five days after surgery. The rats were cared for by a veterinarian in the Center for Laboratory Animals. The surgical site and the activities of the rats were observed daily.

### 4.3. Shockwave Therapy

The rats in the shockwave (SW) group received SW (Figure 1) at the seventh week when OA had been formed in rat knee. The source of the shockwave was a DUOLITH SD1 (STORZ MEDICAL AG, Tägerwilen, Switzerland). The focused shockwave was applied to the medial tibia condyle of the left knee at 0.5 cm below the joint line and 0.5 cm from the medial skin surface under ultrasound guidance [25]. Each rat knee received 800 impulses of shockwave at 0.25 mJ/mm^2^ energy flux density and 4 Hz in a single session. After treatment, all rats were returned to their housing cage for health care and regular daily observations (Figure 1A).

### 4.4. Isolation of Rat Autologous Adipose-Derived Mesenchymal Stem Cells

The rat autologous adipose-derived mesenchymal stem cells were prepared as described in a previous study [25]. The adipose tissue (3 mL for each 1 g of tissue) from the abdomen of rat was minced into small pieces and digested with 0.1% collagenase type 1 (GIBCO, Thermo Fisher Scientific, Inc., Waltham, MA, USA) at 37 °C with shaking for 2 h. After digestion, an equal volume of Dulbecco’s Modified Eagle Medium (DMEM) containing 10% fetal bovine serum (FBS; Gibco; Thermo Fisher Scientific, Inc., Waltham, MA, USA) was added and mixed well. The cell suspension was filtered through a 100-μm filter (BD Biosciences, Franklin Lakes, NJ, USA) for the removal of the solid aggregates. The sample was subsequently centrifuged at 2000 rpm for 5 min at room temperature, and the centrifugation step was repeated. The pellet was re-suspended in 1 mL of red blood cell lysis buffer (Promega, Madison, WI, USA) to lyse red blood cells and incubated for 10 min, then washed with 10 mL of PBS with a 1% antibiotic–antimycotic mixture (Sigma-Aldrich, St. Louis, MO, USA), followed by centrifugation at 2000 rpm for 5 min. The cell pellet was re-suspended in complete medium (DMEM with 20% FBS and 1% antibiotic-antimycotic solution) in a 10 cm^2^ culture dish and incubated in a humidified atmosphere of 5% CO_2_ at 37 °C.

### 4.5. ADSCs Cell Morphology, Phenotype Identification, and Intra-Articular Injection

Homogenously spindle-shaped ADSCs were observed after three to five passages in cultures. The specific surface markers of ADSCs were characterized using flow cytometry. The ADSCs were detached with 1× Trypsin-EDTA in phosphate-buffered saline (PBS) and incubated with a specific antibody (Sigma-Aldrich, St. Louis, MO, USA) conjugated with fluorescein isothiocyanate (FITC) or phycoerythrin (PE) against the indicated markers CD29, CD45, CD90, CD106, RT1a, and RT1b. Subsequently, the cell preparations were analyzed using a BD LSRII flow cytometer (BD, Franklin Lakes, NJ, USA). The ADSCs were intra-articularly injected into the rat knee with 1 × PBS containing 100μL of 1 × 10^6^ cells or 200 μL of 2 × 10^6^ cells under ultrasound guidance (Figure 1A and Appendix A). After 30 min, SW treatment was applied to the rat knee for treatment of OA (Figure 1A and Appendix A).

### 4.6. Micro-CT Analysis

The harvested left lower limb of rat was subjected to micro-CT scanning (SkyScan, 1076, Kartuizersweg 3B 2550 Kontich, Belgium) with isotropic voxel size of 36  ×  36  ×  36 μm^3^ and analysis. The bone surface/volume ratio, trabecular thickness, and trabecular number were measured and computer analyzed. Image reconstruction was performed, and a series of planar transverse and sagittal gray images were generated using NRecon software (Version: 1.7.0.4, Bruker, Luxembourg, Belgium). The region of interest (ROI) of the bone was selected by a semiautomatic contouring method and segmented into binary images using the Skyscan CT-analyzer program. Examination of the trabecular bone included analysis of the trabecular volume fraction (BV/VT), trabecular thickness, and trabecular number.

### 4.7. Histopathological Examination and OARSI Score

The left knee specimens underwent histopathological examination. Left knees were fixed with 4% PBS-buffered formaldehyde at 4 °C for two days, then decalcified in 10% PBS-buffered EDTA at 25 °C for one month. Decalcified knees were then embedded in paraffin wax and dissected into 5-μm-thick sections. The samples were stained with hematoxylin-eosin, and safranin-O (Sigma-Aldrich, St. Louis, MO, USA), and the articular cartilage of knees was graded histologically using Osteoarthritis Research Society International (OARSI) scores for assessments of pathological cartilage structure. The level of damage to the degenerative cartilage was measured using the OARSI cartilage OA grading system by safranin-O staining [46]. The scores were obtained on a 0-to-24 scale by multiplying the index of grades with stages (Appendix A) [25].

### 4.8. Synovitis Scoring

The specimens from the left knees of rats were stained with haematoxylin–eosin to evaluate synovitis score, which included analysis of the thickening of the synovial lining, cellular hyperplasia, and cellular infiltration into joint cavity and synovium. Three features of chronic synovitis described the score rankings, and the overall grades were defined as follows: (1) score of 0 to 1: no synovitis; (2) score of 2 to 4: low-grade synovitis; (3) score of 5 to 9: high-grade synovitis [25].

### 4.9. Immunohistochemical Analysis

The left knee specimens were further analyzed via immunohistochemical analysis of TSG-6, PRG-4, TIMP-1, Type II collagen, BMP-2, and BMP-6. The harvested cartilage and bone specimens were fixed in 4% PBS-buffered formaldehyde for 48 h and decalcified in PBS-buffered 10% EDTA solution. Decalcified tissues were embedded in paraffin wax, and the specimens were cut longitudinally into 5-μm thick sections and transferred to poly-lysine-coated slides. Sections of the specimens were immunostained with specific reagents for anti-rat TSG-6, PRG-4, TIMP-1, Type II collagen, BMP-2, and BMP-6 (Abcam Inc., Cambridge, Massachusetts, USA) to identify the markers present during cartilage degeneration in the rat knees. The immunoreactivity of specimens was demonstrated using a horseradish peroxidase (HRP)-3′-, 3′-diaminobenzidine (DAB) cell and tissue staining kit (R & D Systems, Inc. Minneapolis, MN, USA). The immunoreactivity was quantified from five areas in three sections of the same specimen using a Zeiss Axioskop 2 plus microscope (Carl Zeiss, Gottingen, Germany). All images of each specimen were captured using a Cool CCD camera (SNAP-Pro c.f. Digital kit; Media Cybernetics, Silver Spring, MD, USA). Images were analyzed using Image-Pro^®^ Plus image analysis software (Media Cybernetics, Silver Spring, MD, USA). The percentage of immuno-labeled positive cells over the total cells in each area was calculated, and the average of each specimen was taken as the result.

### 4.10. Statistical Analysis

The software of SPSS ver. 17.0 (SPSS Inc., Chicago, IL, USA) was used in statistical analysis. The data from treatments within the same group were compared statistically using a paired t test. Data of treatment groups and the Sham group were compared statistically using the Mann–Whitney U test. The data of treatment groups were compared statistically using the Chi square test. Statistical significance was set at *p* < 0.05, *p* < 0.01, and *p* < 0.001.

## 5. Conclusions

Treatment with ADSCs had a dose-dependent effect, and a high dose was more effective for the treatment of OA. ADSCs and SW therapy resulted in recovery of the destruction of hyaline cartilage and significantly reduced synovitis. In addition, ADSCs and SW therapy modulated the anabolism of articular cartilage in terms of the expression of TSG-6, PRG-4, TIMP-1, and type II collagen. Further, inflammation-induced BMP-2 and BMP-6 were modulated by ADSCs and SW therapy, resulting in cartilage regeneration in OA. Finally, high-dose ADSCs combined with SW therapy was demonstrated to be a highly efficient therapy, and synergistically improved the pathological changes present in OA for treatment of the condition.

## Figures and Tables

**Figure 1 pharmaceuticals-14-00318-f001:**
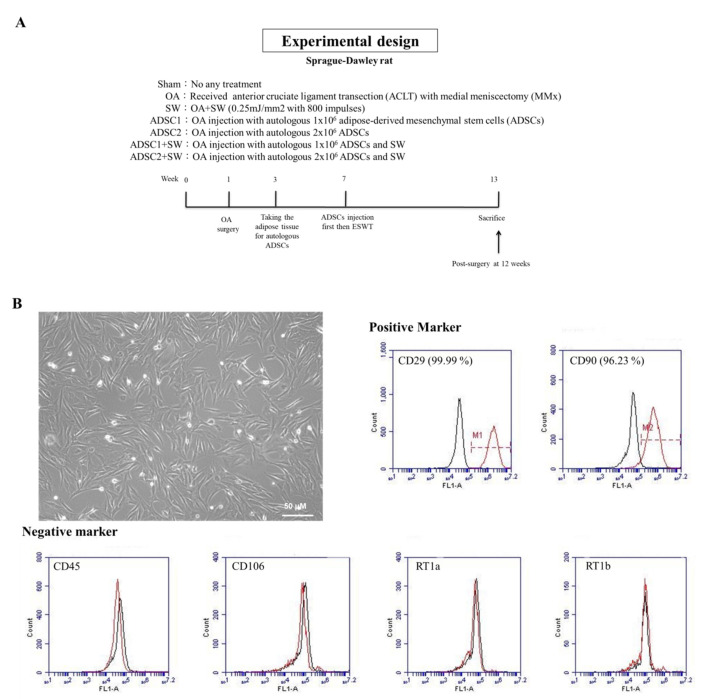
The study design and identification of autologous adipose-derived mesenchymal stem cells (ADSCs). (**A**) Seven groups were included: Sham, osteoarthritis (OA), shockwave therapy (SW), OA injection with autologous 1 × 10^6^ ADSCs (ADSC1), OA injection with autologous 2 × 10^6^ ADSCs (ADSC2), OA injection with 1 × 10^6^ ADSCs and SW (ADSC1 + SW), and OA injection with 2 × 10^6^ ADSCs and SW (ADSC2 + SW). The animals were sacrificed post-surgery at 12 weeks. (**B**) Morphology of cultured autologous ADSCs. White bar = 50 μm. The cell surface makers of autologous ADSCs were analyzed. Positive markers were CD29 (99.99%) and CD 90 (96.23%); negative markers were CD45, CD106, RT1a, and RT1b. The expression percentages of marker were calculated (red line) and isotype controls were shown (black line).

**Figure 2 pharmaceuticals-14-00318-f002:**
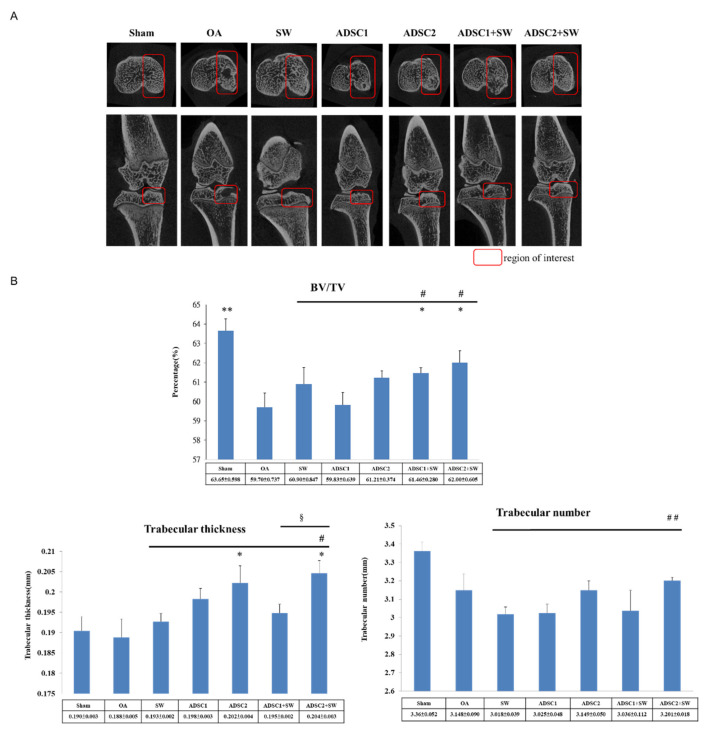
The analysis of Micro-CT scans of the left rat knee after treatment. (**A**) Photomicrographs of the knee from the transverse (upwards) view and sagittal (downwards) view of medial tibia. The region of interest (ROI) for each group is marked (red box). (**B**) The data illustrate the changes in the trabecular bone volume fraction (BV/TV), trabecular thickness, and trabecular number in the ROI. * *p* < 0.05 and ** *p* < 0.01 as compared with the OA group. ^#^
*p* < 0.05 and ^##^
*p* < 0.01 as compared with the SW group. ^§^
*p* < 0.05 as compared with the ADSC1 + SW and ADSC2 + SW groups. *n* = 6 rats in each group.

**Figure 3 pharmaceuticals-14-00318-f003:**
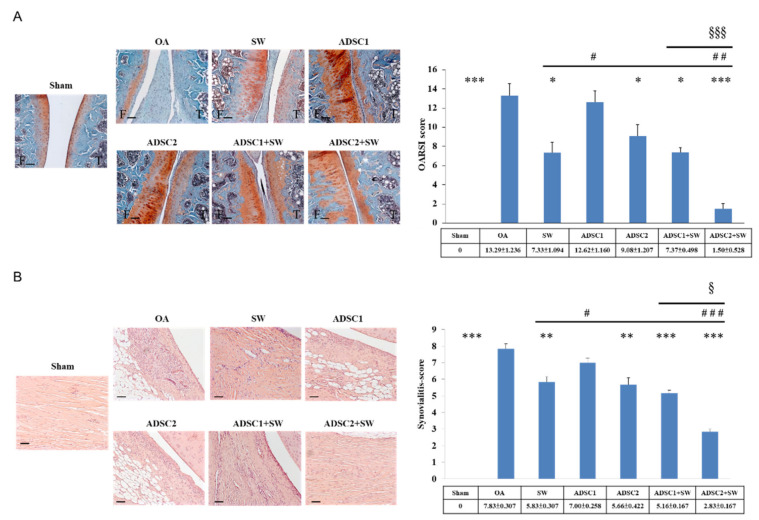
The histological changes of the articular cartilage and synovium membrane in knee OA after treatment. (**A**) Damage to and recovery of the articular cartilage in each group as measured by safarine-O staining and the OARSI score. The images were at 100× magnification. The scale bar was 100 μm. (**B**) The inflammatory synovial membrane observed by hematoxylin-eosin staining at 200× magnification. The scale bar was 50 μm. The synovitis score was calculated for each group. * *p* < 0.05, ** *p* < 0.01, and *** *p* < 0.001 as compared with the OA group. ^#^
*p* < 0.05, ^##^
*p*< 0.01, and ^###^
*p* < 0.001 as compared with the SW group. ^§^
*p* < 0.05 and ^§§§^
*p* < 0.001 as compared with the ADSC1 + SW and ADSC2 + SW groups. *n* = 6 rats in each group.

**Figure 4 pharmaceuticals-14-00318-f004:**
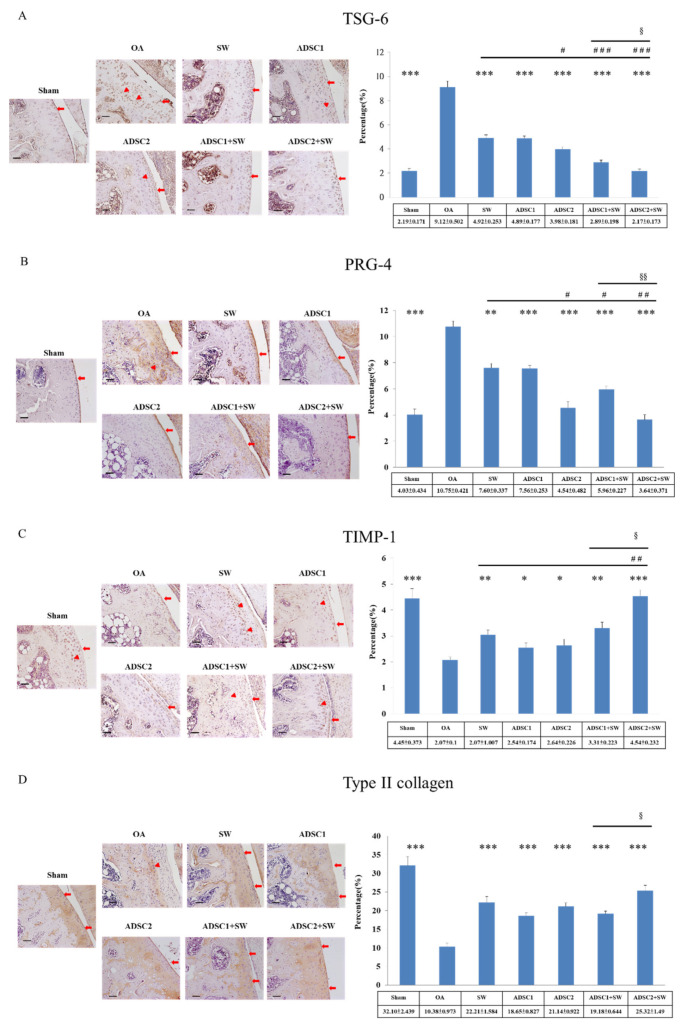
The changes in extracellular matrix factors in the articular cartilage of the OA rat knee after treatment. (**A**) TSG-6, (**B**) PRG-4, (**C**) TIMP-1, and (**D**) Type II collagen were measured in the articular cartilage of the tibia after treatments. The expression percentages were displayed on the left of the images. * *p* < 0.05, ** *p* < 0.01, and *** *p* < 0.001 as compared with the OA group. ^#^
*p* < 0.05, ^##^
*p*< 0.01, and ^###^
*p* < 0.001 as compared with the SW group. ^§^
*p* < 0.05, and ^§§^
*p* < 0.01 as compared with ADSC1 + SW and ADSC2 + SW groups. The images were shown at 200× magnification. The scale bar was 50 μm. The triangle was indicated deep region, and the arrowhead was indicated cartilage surface. *n* = 6 rats in each group.

**Figure 5 pharmaceuticals-14-00318-f005:**
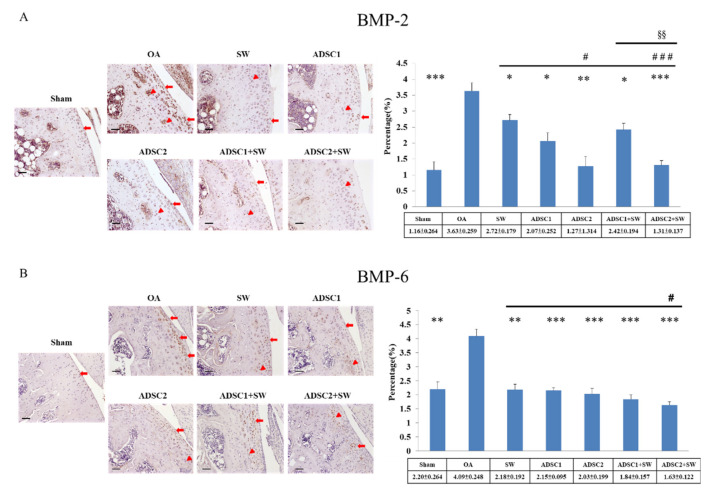
The levels of inflammation-induced BMP-2 and BMP-6 in the articular cartilage of OA rat knee after treatment. (**A**) BMP-2 and (**B**) BMP-6 as measured by immunohistochemical staining in the articular cartilage of the tibia after treatment. The expression percentages were shown on the left of the images. * *p* < 0.05, ** *p* < 0.01, and *** *p* < 0.001 as compared with the OA group. ^#^
*p* < 0.05 and ^###^
*p* < 0.001 as compared with the SW group. ^§§^
*p* < 0.01 as compared with the ADSC1 + SW and ADSC2 + SW groups. The images were shown at 200× magnification. The scale bar was 50 μm. The triangle was indicated deep region, and the arrowhead was indicated cartilage surface. *n* = 6 rats in each group.

**Figure 6 pharmaceuticals-14-00318-f006:**
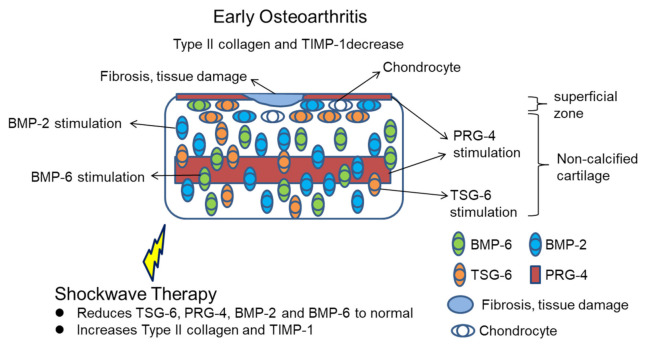
Schematic representation of shockwave therapy modulates the expression of TSG-6, PRG-4, BMP-2, BMP-6, Type II collagen, and TIMP-1 in early OA.

## Data Availability

The data presented in this study are available in the main text and Appendix A.

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
