# Peer review of "Autologous Adipose-Derived Mesenchymal Stem Cells Combined with Shockwave Therapy Synergistically Ameliorates the Osteoarthritic Pathological Factors in Knee Joint"

_pharmaceuticals, 2021, doi:10.3390/ph14040318_

Round 1
Reviewer 1 Report
Dear Authors,
Thank you for submitting this paper; it shows some great potential but it must be improved.
The scoring provided can change dramatically if the points raised in the attached document get addressed.
I hope the suggestions will be useful.
All the best with the review and the re-submission.
Best regards

Author Response
Manuscript review
Autologous adipose-derived mesenchymal stem cells combined with shockwave therapy synergistically ameliorates the osteo-arthritic pathological factors in knee joint By Cheng et al.
Brief summary
The paper presents some very interesting and current results, with great potential translation for tackling OA. In particular, the authors try to pinpoint the best combination therapy consisting of a concomitant administration of adipose-derived Mesenchymal Stem/Stromal Cells (ADMSCs) at different, increasing doses, with shockwave (SW) therapy, ina a rat model of OA.
The paper is interesting as it provides insights into the potential areas of application of a non-invasive biophysical-based regimen of treatment, which is an up-and-coming approach. The outcomes measured are pertinent; particularly good is the incorporation of a current technique like microCT scanning, used along with classic histological and immunohistochemical analysis protocols. The findings support the potential for combining ADMSCS with SW therapy as an avenue to improve osteochondral defects in OA and modulating the pro-inflammatory environment to create a milieu that can favour tissue regeneration.
Broad comments
Whilst potentially very good in terms of study design and outcomes, the study has some major flaws.
First of all, it needs a deep review to improve the English that is unfortunately not good in all the sections. The authors need to address these points as a major concern as it goes at detriment of the paper itself.
Response: Thank you so much for your comments and suggestion and the paper has been improved by the English editing team of OxBioSci England and the certificate is enclosed.
The introduction must be written more eloquently. The level of information provided is rather basic and often consisting of generic statements or sentences that need toning down (for example, it is claimed that ADMSCs are a mainstream therapy in OA, and this is not the case. Their applicability in the clinical setting is true, but the actual long term clinical benefit of ADMSCs is yet to be fully demonstrated). Although summarized by covering some main points, the pathophysiology of OA needs to be written better, by highlighting the role and cross-talking of the main cellular components involved (mainly chondrocytes, osteoblasts/osteocytes and, indeed the synovial fibroblasts). Another point that is completely omitted from the introduction is the one of obesity being main risk factor and contributor to OA, not only from a biomechanical point of view, but also in terms of the adipose tissue being an organ responsible for secreting pro-inflammatory molecules and adipokines with a role in driving OA. Finally, the authors should introduce SW better, especially being a key novelty element of the study. The statement that SW therapy acts as a mechanical stimulus to induce the plethora of effect listed should be explained better-what are the actual mechanistic effects? What are the candidate cellular pathways that are triggered? The ‘listed’ effects of mechanotransduction need clarification and a better context.
Response: We revised the section of Introduction. All the suggestions of the reviewer were considered and wrote to the Introduction.
The aim of the study needs also rephrasing: the authors didn’t do an ‘OA treatment’ but rather they tested a potential treatment in an OA model of rat (and this should be specified).
Response: Thanks suggestion. We specified the description of the “OA treatment” to “OA treatment in the rat model”.
The result section must be improved both in the text and in the figures. In the text there is a limited use of numerical references to the actual results obtained. Things are described as ‘higher’ or ‘better’ or ‘significantly increased/decreased’, but as a standard one should write numerical outcomes, and then, when these are compared and differences between treatment groups are observed, it is recommendable to indicate a percent or fold change, and indicate also the p values in bracket.
Response: We agree with the reviewer’s suggestion. We improved the manuscript, figure and figure legend. We also verified the sentence and indicated the P values in the section of Results.
Histological section microphotographs all miss a scale bar. Why? This needs to be added. In the figures showing histological sections, the areas of interest should be high lighted with insets, dotted lines, pointing arrows, and other similar means to show what the reader should focus on. This will increase clarity for a reader who is not an expert in musculoskeletal tissue microanatomy/histology. Also, the architecture of the histological sections should be better described.
Response: Thanks the suggestion. The loss of pointing arrows and magnification of the images were verified in the manuscript and figure legends.
I suggest to include a table showing the OARSI histological grading criteria (perhaps as part of the supplementary materials?), for a better clarity of how the scores were calculated.
Response: Thanks the suggestion. We created a table for the OARSI score in the supplementary materials as the Supplemental Table 1 as bellowing.
Supplemental Table 1. OARSI analysis after treatments.
|
OARSI Analysis |
Sham |
OA |
SW |
ADSC1 |
ADSC2 |
ADSC1+SW |
ADSC2+SW |
|
grade |
0.00±0.00 |
4.96±0.12 |
2.88±0.22 |
4.71±0.10 |
4.13±0.21 |
3.42±0.08 |
1.92±0.31 |
|
stage |
0.00±0.00 |
2.67±0.21 |
2.50±0.22 |
2.67±0.21 |
2.17±0.17 |
2.17±0.17 |
0.67±0.21 |
|
Score=grade × stage |
0.00±0.00 |
13.29±1.24 |
7.33±1.09 |
12.63±1.16 |
9.08±1.21 |
7.38±0.50 |
1.50±0.53 |
The discussion needs a significant improvement: one of the main issues is that the unfortunately low level of written English makes it difficult to ascertain the level of self-critical awareness. There is a general impression that the actual discussion does not discuss or evaluate the results. The explanation/discussion of the microCT scanning data is omitted, when it should be evaluated. The same goes for the rest of the results; the discussion seems more of a list/explanation of what we know of key OA markers analysed, but without an actual reference or explanation to how the SW therapy, alone or in combination with ADMSCs, does modulate histological/microanatomical features.
Response: Thanks the suggestion. We elucidated the results of micro-CT analysis in the second paragraph of the Discussion.
Another omission in the discussion is the contextualization of the results obtained in relations to the functional and mechanistic effects of SW. These are listed in the introduction and left poorly explained in there; here in the discussion the lack of evaluation, for example of how the hinted ‘mechanotransduction’ of SW may have changed the microarchitectural quality of the osteochondral defects repair.
Likewise, there a vague reference to the capacity of ADMSCs to have likely contributed in modulating the inflammatory microenvironment, and how, in turn, this was reflected in the histological/ICC observations.
Response: Thank you so much for your comments and suggestion. We described the immunomodulation of ADSCs and SW in the introduction and discussion as well as the mechanical force to influence the expression of cartilage extracellular matrix proteins after OA treatment.
The methods section requires mainly a review of the English, but seems comprehensive otherwise. As part of this review, I am omitting line-by-line specific comments, as I feel that after a review of the English it may be better understood what the authors actually want to say.
Response: Thank you so much for your comments and suggestion and the paper has been improved by the English editing team of OxBioSci England.
I am just making one reference to line 258- the cartilage is defined as ‘inflammatory’. I am not sure if this is a correct use of the terminology. The cartilage can be negatively modulated by a pro-inflammatory environment but it is not correct to serine it as ‘inflammatory’. These types of confusions with the terminology need to be addressed throughout the manuscript and be re-evaluated as part of the resubmission.
Response: Thank you so much for your comments and suggestion. We revised the manuscript and verified the “inflammatory cartilage” to “cartilage with inflammation”.
In conclusion, the paper has a great potential and the data presented are solid, but need a better contextualization and explanation.

Reviewer 2 Report
The authors studied the effect of autologous adipose-derived mesenchymal stem cells (ADSCs) and shockwave (800 impulses of 349 shockwave at 0.25 mJ/mm2 energy flux density) on an animal model of osteoarthritis (OA). Transplantation of high dose autologous ADSCs combined with SW (ADSC2+SW) could synergistically ameliorate the process of inflammation and the pathological changes of OA. Significant improvements in the bone remodeling, protection of the articular cartilage, synovitis were observed in the treatment group. In addition, extra cellular matrix factors in the articular cartilage such as TSG-6, PRG-4, BMP-2 and BMP-6 were reduced, whereas TIMP-1 and type II collagen levels increased in the treatment groups. ADSCs treatment showed dose-dependent effect and the effect of ADSCs appeared to be greater than that of SW, especially in the case of BMP-2 decrease. Overall, the study was well-designed and the results obtained support authors’ conclusion. BUT, a lot of sloppiness can be found throughout the manuscript. Moreover, it’s hard to read text and needs extensive English editing. It is strongly recommended that the manuscrpt be edited by a native speaker of English. The following points must be addressed as well.
Points
- The Discussion can have a figure summarizing the text, which will help the reader greatly.
- Statistical software or the method used needs to be described in more detail.
- The following typos or errors must be corrected. This is just a part of the whole case. Please check the manuscript thoroughly.
line 68; chomdroprotection
line 87; along
line 136; highly expression
line 142; beat
line 147: collage
line 163 should be removed.
line 198: synnovium
Author Response
Comments and Suggestions for Authors
The authors studied the effect of autologous adipose-derived mesenchymal stem cells (ADSCs) and shockwave (800 impulses of 349 shockwave at 0.25 mJ/mm2 energy flux density) on an animal model of osteoarthritis (OA). Transplantation of high dose autologous ADSCs combined with SW (ADSC2+SW) could synergistically ameliorate the process of inflammation and the pathological changes of OA. Significant improvements in the bone remodeling, protection of the articular cartilage, synovitis were observed in the treatment group. In addition, extra cellular matrix factors in the articular cartilage such as TSG-6, PRG-4, BMP-2 and BMP-6 were reduced, whereas TIMP-1 and type II collagen levels increased in the treatment groups. ADSCs treatment showed dose-dependent effect and the effect of ADSCs appeared to be greater than that of SW, especially in the case of BMP-2 decrease.
Overall, the study was well-designed and the results obtained support authors’ conclusion. BUT, a lot of sloppiness can be found throughout the manuscript. Moreover, it’s hard to read text and needs extensive English editing. It is strongly recommended that the manuscript be edited by a native speaker of English. The following points must be addressed as well.
Response: Thank you very much. The manuscript has carefully improved the grammar and readability by the English editing team of OxBioSci England and with the certificate.
Points
The Discussion can have a figure summarizing the text, which will help the reader greatly.
Response: We would like to thank the suggestion and added the Figure 6 to summarizing the results in the manuscript as bellowing.
Figure 6. Schematic representation of shockwave therapy modulates the expression of TSG-6, PRG-4, BMP-2, BMP-6, Type II collagen and TIMP-1 in early OA.
Statistical software or the method used needs to be described in more detail.
Response: We appreciate for the reviewer's recommendation and detail described the statistical software to use in the experiment.
The following typos or errors must be corrected. This is just a part of the whole case.
Please check the manuscript thoroughly.
line 68; chomdroprotection
line 87; along
line 136; highly expression
line 142; beat
line 147: collage
line 163 should be removed.
line 198: synnovium
Response: Thanks for the reviewer’s reminding. We carefully verified all the typos in the manuscript.

Round 2
Reviewer 1 Report
Dear Authors,
Thanks for the major effort in revising the manuscript.
I believe the figure presentation has been in part improved, but not consistently in all the figures presented. You still miss the scale bar in all the histological microphotographs. Arrows7insets/pointers were added only to some figures, which detracts from clarity. The discussion, in my opinion, is still mostly descriptive rather than evaluative but is much improved. I hope that with the help of the editors you can polish the final detail and make all the figures consistent. I look forward to seeing the published version.
Author Response
Thanks for the major effort in revising the manuscript.
I believe the figure presentation has been in part improved, but not consistently in all the figures presented. You still miss the scale bar in all the histological microphotographs. Arrows7insets/pointers were added only to some figures, which detracts from clarity. The discussion, in my opinion, is still mostly descriptive rather than evaluative but is much improved. I hope that with the help of the editors you can polish the final detail and make all the figures consistent. I look forward to seeing the published version.
Response: Thank you so much for your comments and suggestions. The scale bars, triangle, and arrow heads were added in the photographs. We are grateful to the reviewer for insightful comments on the manuscript.
